# A Comprehensive Security Architecture for Information Management throughout the Lifecycle of IoT Products

**DOI:** 10.3390/s23063236

**Published:** 2023-03-18

**Authors:** Narges Yousefnezhad, Avleen Malhi, Tuomas Keyriläinen, Kary Främling

**Affiliations:** 1Department of Computer Science, Aalto University, 02150 Espoo, Finland; 2Department of Computing and Informatics, Bournemouth University, Poole BH12 5BB, UK; 3Department of Computing Science, Umeå University, 90187 Umeå, Sweden

**Keywords:** Internet of things (IoT), information management, security architecture, product lifecycle information management (PLIM), identity and access management (IAM)

## Abstract

The Internet of things (IoT) is expected to have an impact on business and the world at large in a way comparable to the Internet itself. An IoT product is a physical product with an associated virtual counterpart connected to the internet with computational as well as communication capabilities. The possibility to collect information from internet-connected products and sensors gives unprecedented possibilities to improve and optimize product use and maintenance. Virtual counterpart and digital twin (DT) concepts have been proposed as a solution for providing the necessary information management throughout the whole product lifecycle, which we here call product lifecycle information management (PLIM). Security in these systems is imperative due to the multiple ways in which opponents can attack the system during the whole lifecycle of an IoT product. To address this need, the current study proposes a security architecture for the IoT, taking into particular consideration the requirements of PLIM. The security architecture has been designed for the Open Messaging Interface (O-MI) and Open Data Format (O-DF) standards for the IoT and product lifecycle management (PLM) but it is also applicable to other IoT and PLIM architectures. The proposed security architecture is capable of hindering unauthorized access to information and restricts access levels based on user roles and permissions. Based on our findings, the proposed security architecture is the first security model for PLIM to integrate and coordinate the IoT ecosystem, by dividing the security approaches into two domains: user client and product domain. The security architecture has been deployed in smart city use cases in three different European cities, Helsinki, Lyon, and Brussels, to validate the security metrics in the proposed approach. Our analysis shows that the proposed security architecture can easily integrate the security requirements of both clients and products providing solutions for them as demonstrated in the implemented use cases.

## 1. Introduction

The idea of having online access to product information throughout the product lifecycle was introduced as early as 2003 [1], i.e., before the concept of IoT started to gain popularity in around 2010. Due to the excessive attention it received, the IoT has grown faster than expected and it is now playing an outstanding role in everyday life. As the IoT expands, an ever-increasing number of devices and systems will become interconnected. At the same time, all information transfers between such systems require to be managed throughout the whole lifecycle of those IoT-connected products. Therefore, the IoT must provide the necessary systems of system-level interoperability among systems that generate, store, or employ lifecycle-related information.

Information management during the lifecycle of the product is an important challenge in IoT which is hindered by the structure of the interaction among various applications. In other words, product information is blocked in a related application (blue arrows in Figure 1) unaccompanied by the possibility of sharing with other applications.

To share the information through the whole lifecycle and across the whole spectrum of lifecycles, such vertical interactions must be replaced by information loops (black arrows in Figure 1) enabled by common, open, and trustworthy information exchange standards. This concept is referred to as product lifecycle information management (PLIM), as initially proposed in [2]. PLIM manages the whole lifecycle enabled by IoT interoperability, where information from one lifecycle phase affects processes and decision-making in the other phases. PLIM extends the PLM concept to comprise detailed information both about individual product instances (i.e., physical products) and their adoption in the middle-of-life (MoL) and end-of-life (EoL) lifecycle phases [3,4].

### 1.1. Motivation

PLIM aims to provide control over the information flow to know how the information is used, with whom it is shared, and what actions are applied to it to manage the overall lifecycle. Interconnecting objects with each other (i.e., IoT) within PLIM requires structured and auditable protocols to manage the trust and security of exchanged information. The quantity of data exchanged among the entities and the global distribution of entities impose data sharing and security challenges. It is difficult to ensure information flow and security at the same time. The main challenge in PLIM systems is the threat of shared data being compromised. Hence, it is important that the information be securely protected and shared in PLIM systems. On the other hand, since PLIM presents a multifunctional concept at a higher level compared to IoT, PLIM also introduces new security considerations. There has been extensive research covering various limitations of smart products, including the massive quantity of data generated in the network, the heterogeneity of the data, and dynamic changes in the network. However, PLIM security is entirely missing from the state of the art which is a great gap in the literature. Given this concern, it is essential to study various aspects of security in PLIM.

Information security and its management in the IoT ecosystem should be improved by a lifecycle approach where security is managed from the device manufacturing phase all the way to the disposal of the device [5]. Addressing security problems in PLIM also requires a comprehensive security framework that can be applied throughout the entire lifecycle. Accordingly, this paper addresses the issue by providing an integrated identity and access management (IAM) system which provides interoperability between various vendors. Furthermore, to implement such a management system, we propose a security architecture with an ongoing process over the lifecycle. On the other hand, it would be beneficial if such architecture was combined with the digital twin (DT) concept and its elements, including a physical space, a virtual space, and their connections, as implemented in a smart campus (see more information in [6]), because managing lifecycle information and access to it through a DT makes security easier to establish, even though the actual information might be distributed over many systems.

### 1.2. Key Contributions

The main contributions of this paper are summarized as follows:To the best of our knowledge, this research is the first study focusing on the security of PLIM and the importance of IoT security through the lifecycle.A new security architecture for PLIM and the IoT is proposed that manages the identity and access levels of users covering the entire product lifecycle.The used integrated IAM system is flexible and modular enough to be installed on diverse organizational structures and provides a secure connection among various vendors.

### 1.3. Structure of Paper

The article is organized as follows. First, we present relevant interoperability standards, identity management tools exploited in our approach, and a comparison with other IAM platforms in Section 2. In Section 3, we review the literature on the security of the IoT and PLIM. Then, a detailed explanation of the proposed architecture is provided in Section 6. Finally, Section 7 analyzes and discusses the security considerations which lead to the paper’s conclusion with our key findings in Section 9.

## 2. Background

This section details the related background concepts of the O-MI/O-DF standards and identity management, and finally the comparison of our IAM presented for users with other IAM tools on the market.

### 2.1. O-MI and O-DF Standards

Many protocols have been designed for machine-to-machine communication, particularly in the IoT environment. However, they are not sufficiently generic or flexible to be able to support the diverse organizational requirements and structures and enable interoperability between various vendors. To fulfill these requirements, the Open Group consortium published two standards, called Open Messaging Interface (O-MI) and Open Data Format (O-DF) messaging. Version 1.0 of such standards was published in 2014 and version 2.0 in 2019. O-MI and O-DF for IoT were proposed with the same purpose as HTTP and HTML for the web. For PLIM applications, O-MI establishes a communication procedure between products and distributed information systems that consume and publish information on a real-time basis. Considering a simple ontology, O-DF is characterized as an extensible XML schema, to display the payload in IoT applications. Similarly, data structures of object-oriented programming are intentionally adopted to define O-DF. It offers a hierarchical structure including a single top element (i.e., objects) and several subelements. The subelements might contain two types of subelements: properties (i.e., InfoItem) and other subelements (i.e., object) [7,8].

### 2.2. Identity Management

Before introducing the concept of IAM in the IoT environment, another related and general concept referred to as identity management or IdM is presented in this section. IdM refers to the protection of a device profile while also managing and securing access to information and resources. IdM approaches can be employed to define the identity of an entity, for instance, person, thing, or device, to store the related information for entities, to make such information available through an interface, and to manage the relationships among resources and other entities [9]. Various protocols and technologies can be applied in the IdM system:Local authentication: having a server- or service-specific user database is still a common approach for many systems.Kerberos: it is an authentication method based on the idea of a federation of servers. It suffers from some drawbacks; for example, all clocks should be synchronized, it does not handle user data connection to the service, and it does not support external authentication.SAML 2.0 and Shibboleth: SAML 2.0 is an XML-based protocol that adopts security tokens to exchange information about authentication and authorization identities between security domains [10]. Shibboleth is an open-source implementation of SAML 2.0, accessed 1 November 2022, see www.shibboleth.net.OAuth 2.0: It is a delegation-based authorization technique with limited access to user accounts. There are four rules in this method, including resource owner, client, resource server, and authorization server. Different implementations for various domains and insecure static user/pass constitute the main issues in this solution.OpenID Connect: It is a distributed authentication consisting of OpenID providers and an acceptor. It provides a single-sign-on user experience. However, it has some drawbacks. For instance, for authentication, the server needs to be trustful and accessible or its application will be limited to browsers only.

Furthermore, there are some IdM or authentication methods devoted to IoT platforms. According to Mahalle et al. [11], three types of IdM models exist for IoT: user-centric (e.g., Windows Card-Space and OpenID), device-centric (e.g., Higgins and OAuth), and hybrid (identity of both user and device; e.g., Liberty Alliance). Thuan et al. [12] proposed a user-centric IdM model based on the combination of user identity (ID), device ID, and their relations. This system was provided by subsystems for a device, a service, and an identity provider (IdP) and applied a user ID for device authentication and authorization. In addition to a globally unique ID created by the manufacturer, each device in this model had several IDs which could be defined as a combination of an identifier of the IdP, an identifier of an organizational domain, an identifier of a device, and an identifier of a user (i.e., DeviceID: idpID/domIDPart/devIDPart/userIDPart). Horrow and Sardana [13] also presented an IdM framework for cloud-based IoT.

### 2.3. Comparison with Other IAM Tools

IAM in the IoT is about determining and managing the network users’ roles, access privileges, and the situation where these privileges can be granted (or denied) for the requested users. IAM systems allocate a set of tools and technologies for administrators with which they can change users’ roles, track their activities while creating reports on them, and enforce policies. We compare the IAM tools provided by different services on the market with the IAM used on the user side of our security architecture. As shown in Table 1, all IAM platforms enable grant sharing with granular permissions, while providing single sign-on (SSO) and identity federation, which allows users to securely access resources from different domains without another user account. In addition to the common features among all the IAM platforms, our IAM system provides free access to the IAM services for any user, although other platforms can be used without payment only if the user has already paid for the cloud services provided by that company.

O-MI reference implementation connects to authentication and authorization modules through a configurable REST API, which makes the system modular and flexible. It is also possible to use a third-party authentication system, as it suffices to acquire a user ID for the authorization module to work. An O-MI node can be configured to retrieve cookies, headers, or tokens from the O-MI request and pass them on to the authentication module. Similarly, parts of the authentication response can then be extracted for the authorization module request, which then needs to satisfy the access rules. The system is made even more modular and flexible by the fact that it is open source, which means that adapters or custom features can be designed easily as the developers can see the inner workings or modify the code directly. Another advantage of open source is that it can also be run locally, rather than cloud-only as the other IAM tools compared with here. The cloud might be avoided for reasons such as privacy, control, or vendor lock-in concerns. With the modular and microservice-based architecture of our system, it is also possible to construct a hybrid system where only the required information is stored or processed in the cloud.

## 3. Related Work

As discussed earlier in Section 1, the management of data associated with physical entities during their lifecycle is named PLIM and its security is of paramount importance. However, according to our findings, it is a quite new topic with no security approach particularly assigned to it. Given this concern, to properly configure this section, we discuss the security of concepts that highly correlate with PLIM, including the IoT and the DT. In other words, an intersection of IoT and DT concepts establishes PLIM as a broader concept (see Figure 2).

### 3.1. Security Elements in the IoT

To detect possible vulnerabilities and security threats in IoT systems, it is first necessary to discover the main elements of the IoT. However, the literature includes various interpretations of these elements. According to Burhan et al. [14], the basic elements of IoT are identification, sensing, communication, computation, services, and semantics. The elements of IoT systems can also be divided into four groups: individual devices and smart objects, edge nodes, IoT platforms, and entire clouds [15]. Similarly, such elements might be divided based on the security aspects. For instance, to address major technical security controls over IoT-enabled networks, architecture security elements, from network and device perspectives, can be classified into secure device identifier, secure credential management, secure network access of devices, policy enforcement for devices, or device and system integrity assurance [16].

In the mentioned lists of elements, one important element, the user, is missing. User refers to the most vulnerable element in IoT security [17] and a secure information system with no user-side security will be impractical and considered insecure. To address this issue, Oh and Kim [17] proposed six key elements for IoT systems, the IoT network, cloud, user, attacker, service, and platform, and then investigated their security issues. However, this categorization also missed another important element: the IoT device (any smart and connected device such as a vehicle [18]). To ensure the security of sensors and connected IoT devices, various security issues should be addressed, including physical security [19], key/certificate management [20,21], trust management [22], and compromise detection [23,24,25]. As a consequence, IoT systems require a security architecture that encompasses the most important elements, including device and user.

### 3.2. Security Architecture for the IoT

The concept of security architecture in the IoT can be described differently according to the method, threat target, and application domain. Applying features supported by software-defined networks (SDNs) can be considered a method to introduce a security architecture for the IoT ecosystem. Leveraging SDN, Karmakar et al. [26] presented a security architecture to authenticate IoT devices by a lightweight protocol and to secure network flows by employing fine granular policies. Another SDN-based security architecture was proposed by Olivier et al. [27], where both wired and wireless infrastructures became secure and eventually security reached the network access endpoints (e.g., access switches). nA SDN coupled with blockchain technology builds a decentralized security architecture for the IoT network, which addresses the issue of security attack detection [28].

Security architectures can target various threat objectives including endpoints, edge, and middleware. For instance, Tiburski et al. [29] analyzed the effect of lightweight protocols on the security architecture for IoT middleware since middleware processes a large quantity of data and requires a proper security architecture to ensure the protection of all layers of the system. The same researchers expanded their research by defining a lightweight security architecture for IoT edge devices that incorporated embedded virtualization and trust mechanisms [30]. Targeting end-devices, OSCAR [31] is an object-oriented and producer-consumer security architecture where security is related to the application payload.

Security architectures can also address security challenges in the subcategories of IoT systems. For example, to manage security perspectives in three layers including information, physical, and management, Ning and Liu [32] designed a CPS-based security architecture supporting U2IoT (unit IoT and ubiquitous IoT), a future model of the IoT. Some security architectures facilitate the specific application of IoT. For instance, to support various multimedia applications in the IoT, a media-aware security architecture was presented by Zhou and Chao [33] based on their novel traffic classification and analysis method.

### 3.3. Security in DTs

Since no research has been conducted explicitly on PLIM security, we alternatively reviewed the literature on the closest concepts to PLIM, which are the IoT and the DT. As seen in Figure 2, the IoT partly shares a common scope with both PLIM and DTs; however, a DT covers most of the implications in PLIM. A DT is a virtual replica of a physical entity that enables the information to be handled in a virtual enterprise [6]. By applying a DT, all information requests for a given physical product would be achievable over a single address on the Internet throughout the lifecycle, specifically during the design and service phases. A DT could be considered one application of the IoT which has flourished from rapid advances in the IoT. PLIM supervises the product information throughout the lifecycle. A DT, on the other hand, is known as an approach enabling PLIM, thus combining the two concepts could boost both concepts from various aspects.

Although a DT requires a guaranteed secure connection, a DT itself helps in securing the system. By applying a DT, no external physical access is required to be granted to outsourced firms, which might compromise security. Additionally, security experts are only required to analyze the security aspect over simulated environments, which is much easier than analyzing the physical environments where the setup test scenarios and maintaining test scenarios for security approaches are expensive and time-consuming. For instance, Eckhart and Ekelhart [34] proposed a framework to generate a DT over CPS and represented ways in which such CPS twinning could be leveraged for various security use cases such as privacy, intrusion detection, detecting hardware and software misconfiguration, and secure decommissioning. Additionally, Gehrmann and Gunnarsson [35] presented a security architecture based on a DT replication model, which allowed data sharing and the control of security-critical processes. A cost-effective DT with the ability for security evaluation was also proposed by Bitton et al. [36] to solve the DT challenge of the security analysis of industrial control systems, in order to make a trade-off between budget and fidelity.

On the other hand, by integrating all the information in one place over clouds, a DT attracts more attackers and should be handled with extreme security care. It also has its specific security requirements including access control, software security, DoS resilience, external connection protection, and synchronization security [35] spreading over the lifecycle. Therefore, security should be handled throughout the whole lifecycle from the beginning of life and design phase till the end of life, when the product is decommissioned.

**Conclusive research gaps:** Although PLIM can be considered a combination of IoT and DT concepts, security solutions for such concepts cannot render a perfect security approach for PLIM due to slightly different details on each of these concepts. The IoT manages the information at the digital level and in the operation phase; however, PLIM handles it mainly on physical products and in all lifecycle phases, so an adequate security method should secure the information flow on both physical devices and their virtual counterparts considering the lifecycle phases from the beginning of life, middle of life, then end of life. Additionally, PLIM generally enables a user to access the existing information, while a DT signifies the importance of updated data, thus a security solution supporting both existent and updated data would be the best security solution. In conclusion, all security approaches proposed for the IoT or DT concept are insufficient for PLIM and new efforts are required.

## 4. Research Methodology

This paper follows the same methodology as Andrade et al. [37] and Preidel and Stark [38] since they worked on similar topics, including the security analysis in IoT systems and systems of systems’ lifecycle management. The used research methodology, design research methodology (DRM), was proposed by [39] and consists of four stages: research clarification, descriptive study I, prescriptive study, and descriptive study II. Figure 3 presents the methodological stages, together with the main means and outcomes of each stage.

### 4.1. Research Clarification

The first stage of the DRM identifies the most relevant elements of the IoT environment that could be considered in security management and involved with security threats. To achieve this, a systematic literature review (SLR) was built on the topics of security elements, security architecture, and DT security. To improve the establishment of systematic reviews, a four-phase methodology known as the PRISMA Statement was employed. According to PRISMA [40], the phases consisted of identification, screening, eligibility analysis, and inclusion. During the identification phase, resources were selected from Google Scholar and the DBLP bibliography, accessed on 1 November 2022. The screening phase reviewed the titles and abstracts of articles. The eligibility analysis phase was applied to read the full texts of the chosen articles. Finally, in the inclusion phase, selected publications from the eligibility analysis phase were employed to describe the security elements in the IoT, the available security architectures, and the security of the DT and PLIM.

### 4.2. Descriptive Study I

In the second stage, we conducted an in-depth literature review of the security issues over the lifecycle, taking the same elements from stage 1, resulting in a deep understanding of the current state of the art. Additionally, in order to exchange knowledge with experts regarding existing security issues or aspects in practice, the authors participated in several relevant conferences and workshops; additional aspects and issues were provided and added to the comprehensive list of necessary security issues. Finally, to cover the security aspects in all phases of the lifecycle, as seen in Table 2, the security issues were categorized based on the stages and phases of the lifecycle. Consequently, the most significant security issues, considering the security gap were selected to be implemented in the next step.

### 4.3. Prescriptive Study

From the LR, we concluded that a comprehensive security architecture was missing from the literature which could cover the security of both user and sensor devices. Then, a security architecture was proposed that considered both the user and devices as the main security elements of IoT environments. Moreover, the security architecture supported all stages of the product lifecycle. In the prescriptive study, the main security challenges of each security element were implemented which were authentication and access control on the user side and device identification on the device side. The proposed security architecture and implementation details are presented in Section 6 and Section 8.

### 4.4. Descriptive Study II

In the last stage of the research methodology, the proposed security architecture on the device side was validated through an empirical analysis and use cases. The goal of the experiments was to clearly understand the behavior of IoT devices and web clients against security events. For this purpose, two experiments were proposed to simulate security attacks on a small IoT system such as a smart campus. Furthermore, on the user side, the proposed security models were tested on a smart home installation.

## 5. System Model and Problem Definition

This section briefly introduces the system model of the proposed method and defines the problem in more detail.

### 5.1. System Model

IoT devices and users, as two principal elements of the IoT systems, are prone to various cyberattacks such as denial of service, man-in-the-middle, and masquerade attacks. The main reason behind such security vulnerabilities, according to BeyondTrust [41], is insecure communications in IoT environments. A malicious entity can obtain access to sensitive information by intercepting the process of data transmission. To protect against such vulnerability, the connection between users, devices, and servers must be secured, since the data can be transmitted among these three entities. Figure 4 displays the secure communication between the user, device, and server as the system model.

### 5.2. Problem Definition

This paper aimed to guarantee the security of both users and devices. To achieve this, the communication between the user, device, and server must be secured. For this purpose, the legitimacy of the received data should be guaranteed for each legitimate party who has passed the identification process. Each party must also be specified by the particular access privileges to resources. Accordingly, this paper concentrated on three security requirements consisting of identification, authentication, and authorization. These security requirements were defined through a security architecture. Once implemented, they were analyzed through the designed attack and various use cases.

## 6. Proposed Security Architecture

Although the IoT is expected to have a high influence on our lives, it slowly finds its secure way to reality. The reason is that irrespective of the diverse nature of the resources such as hardware, software, and systems, their security requirement must be collected in a single entity, which is an architecture. Thus, to securely deploy an IoT system, a potential security architecture is mandatory.

As discussed in Section 3, security should be considered and managed during the entire lifecycle. For this purpose, a security system is required for PLIM which hinders all unauthorized access to information and restricts the access levels based on user permission [42]. Most of the existing security solutions are restricted to their own association (vendor-specific), indicating a lack of a globally regulated security model which could be employed in any vendor or situation. On the other hand, similarly to the organizational structure which makes the base of a successful organization, a security architecture can be the cornerstone of any security system. To solve the problem, the current section provides a novel security architecture and describes its subsystems, as well as their interactions. Both authentication and authorization (or access control) are provided in the proposed security architecture (see Figure 5). According to this architecture, the access policies can be shared with other users. At the same time it supports identity federation.

This architecture investigates the security breaches during the lifecycle, considering security principles in three layers including device, interoperability, and user. A *secure device* concentrates on device-side security issues such as device identification. *Secure interoperability* is managed by the O-MI server which controls messaging and communication. Including user authentication and authorization, the *secure user* layer facilitates user-side security.

Messaging control through the secure interoperability layer consists of request type checking and O-DF content filtering with the rules provided by the authorization module. This way, each info item is filtered based on the user and whether the request is “write”, “read”, “delete”, or “call”. Communication control is handled mainly by the HTTPS protocol, but in order to prevent abuse via the O-MI callback functionality, callback addresses are required to be the same as the requesting addresses. The *secure interoperability* layer is supported by existing technology and the other two layers remain the main focus of the architecture. Generally speaking, we divided the security in our security architecture into two categories: user-side security and device-side security.

### 6.1. Secure Device (Device-Side Security)

The device-side security in Figure 5, proposed by [43], is a high-level overview of the framework, which determines the device identity by running an automatic classification of IoT devices according to traffic data, sensor measurements, and a classifier model. The system is divided into four main layers and consists of two key modules, including *model management* and *security Management*. *Model Management* initially undertakes the extraction of the most adequate features (dimensions) according to their importance level through ML methods. Based on the extracted features, the ML model of the current device is identified, then *security management* makes the security decision, whether it is authenticated and whether it provides the enforcement support. Additionally, the device-side security system offers two databases (DBs) for managing the data. The first DB, the *data and features DB* consists of dimension names and sensor measurements. Immediately after loading the required dimensions from the first DB, the system elicits the measures of each dimension from the observation and the learned model is stored in the *Model DB*. We refer the reader to [43] for more details.

### 6.2. Secure User (User-Side Security)

A detailed view of our proposed IAM considering the O-MI Node reference implementation security modules and their interaction model is provided in the user-side security part in Figure 5. First, the user adopts an O-MI application, which in this case is the *O-MI WebClient*. The user can log in by going to the log-in form. From the form, the user can select to log in either with a local or an external user account. When logging in with a local account, the request goes to the O-MI *authentication* module, but when logging into an external account, the user is redirected to the *external identity providers* log-in form. The external provider can be either OAuth2.0-based or SAML2.0-based, where the former is handled by means of collaboration between the *O-MI authentication* module and OAuth2 service provider. The latter is handled by the Shibboleth module in the *reverse proxy*. In the *reverse proxy*, SAML2.0 authentication includes the email address as the username in HTTP headers if the authentication was successful otherwise it includes an empty username. This way, the O-MI Node is configured such that it uses either the username directly, or if the *O-MI authentication* module was used, it can also acquire the username from there, by sending the token or cookie to be validated. Finally, the *O-MI node* uses the username to ask the *O-MI authorization* module which requests are allowed or denied for which item and executes the request accordingly.

### 6.3. Components

In order to achieve a clear picture of ways in which the security works internally through the server, Figure 6 represents the internal service components with the main interaction among them. Once data are collected by sensor devices, the *monitoring component* sends the real-time data to the server (known as the *O-MI node*), using the O-MI open standards. The client thus can access the data concerning the systems installed on the client property employing two types of request: *read* and *write*. Whenever the client requires any information from devices, sensors, or databases, they can request an O-MI *Read* operation. On the other hand, the O-MI client adopts the *write* request to write data originating from sensors, events, or other devices to the O-MI node. To communicate with the server, the *handler* in the server first processes incoming/outgoing messages, then other internal components in the server can receive the data. Data can be fetched from the database during *read* requests or can be sent to the database, *agent manager*, and *subscription manager* during *write* requests. The *Agent manager* is responsible for handling the write or read requests of agents and determines the necessity of implementing any proprietary protocols for devices. Event and interval subscriptions are managed by the *subscription manager*, which returns the callbacks including the requested data to the recipients of subscriptions. The O-MI node eventually requests the *authentication* and *authorization* modules to investigate whether each item of the request is authorized to be executed or not. Additionally, the *mentoring component* which is mounted between the O-MI node and data collection components, applies a traffic analyzer such as Pyshark to extract the sensor values and header information from the arrived packets. Such information is stored in a database at the *security* module and is used later for device identification at the *device security* submodule.

## 7. Security Considerations

To ensure a sufficient level of security in the proposed security architecture, this section studies the security considerations through several security practices.

### 7.1. Authentication and Authorization

In O-MI/ O-DF standards, even though the reference implementation guarantees data standardization and robust development, security still poses major concerns. The reference implementation originally owned an IP whitelist mechanism for security where only the trusted IP addresses from the list were able to access the domain and perform the “write” functionality; however, everyone could access the “read” functionality [44]. Hence, an updated security version was designed which divided the authentication and authorization module into two separated individual submodules working concurrently, as shown in Figure 7. The authentication submodule is responsible for registering, authenticating, and logging the users to run sessions. The login options are generally divided into two categories: local and external authentication. In local authentication, the ordinary login process is followed using the username and password which have been stored in the database. The external authentication adopts two different protocols: OAuth 2.0 and SAML 2.0. On the other hand, the authorization submodule includes two segments: the superuser console and the access control module. The administrator console provides administrator functionality for the superuser through a specific user interface. This console is responsible for adding users to groups and managing access policies on the database. Instead, the access control module is responsible for communication with the O-MI node and processing and authorizing the user requests. The implementation details of these submodules are detailed in [44], developed using Python and Scala. The authentication module was developed using the Django web framework along with Akka HTTP for the construction of security models, supporting LDAP directory services and SQL databases. For the token mechanism, JSON Web Tokens (JWT) were adopted. These modules are imperative in making the O-MI node secure by securing APIs while adding extra functionalities such as enhanced access control, user identity, analysis, and monitoring from a multitude of other methods. The detailed authentication process in the three types of login methods are discussed in the following.

#### 7.1.1. Local Authentication

The user opens the browser and on the login web page, decides to choose the local authentication option by entering their login information including their username and password. Then, as seen in Figure 8, the O-MI authentication module verifies the user information in the local database and upon successful authentication, a JWT is returned to the user or a cookie is set on the browser. From thereon, all O-MI requests from the user are sent with the cookie or JWT to the O-MI node server. Then, the server investigates the token validity by asking the O-MI authentication module. Once the token is validated, the access rules should also be fetched from the authorization module and the server will respond to the user based on the rules dedicated to the user. The authentication process ends when the user observes the requested information.

#### 7.1.2. OAuth2.0 Authentication

The OAuth 2.0 integration with the O-MI authentication module is shown in Figure 9 depicting all the steps undertaken for the proposed model. The authorization code OAuth flow is used in this example. As the first step, the user selects an OAuth2 provider as the login method on the O-MI authentication module login page. The browser is then redirected to the selected OAuth2 service provider, where the user can provide the credentials to log in. After the user authorizes the O-MI authentication module to see the email address or other user IDs, the browser is redirected back to the module with the authorization code. The code is then used to trade for the access token as usual, and the user id is acquired. From now on, the O-MI authentication module can continue with the same logic as in local authentication, by setting the cookie or returning the JWT.

#### 7.1.3. SAML2.0 Authentication

The SAML2.0 flow is presented in the sequence diagram of Figure 10. For simplicity, the user, user agent (e.g., a web browser), and the O-MI client application have been combined into the first actor on the left. The other actors are the reverse proxy, SAML identity provider, O-MI node, and O-MI authorization module. The O-MI node contains the data that the app needs to use, and it is set up behind the reverse proxy, which has a Shibboleth module installed for handling the SAML2.0 implementation. To initiate the SAML2.0 authentication, the user needs to select it from any other login option. As a result, the app goes to the URL of the Shibboleth module of the reverse proxy, which in turn redirects to the correct identity provider (IdP). The user can enter the credentials on the IdP login form and the IdP sends the response back to the Shibboleth module via the user browser. Then, Shibboleth can set a session cookie for the user, to recognize the user in further requests. The reverse proxy with the Shibboleth module is configured to include the username in an HTTP header when receiving an O-MI request from authenticated users. The HTTP header is always set or removed by the Shibboleth module to prevent forging. As the next step in the O-MI node, the username is configured to be extracted and directly passed to the O-MI authorization module.

### 7.2. Basic Security Pillars

Information security is principally provided by the CIA triad of confidentiality, integrity, and availability. The principles, which are also known as security pillars, were addressed by the proposed security architecture as follows.

**Confidentiality**. The main security principle is confidentiality where information should not be disclosed to unauthorized individuals. The proposed security architecture provides confidentiality for end-users by means of the authentication module. The confidentiality of network communications is protected through the HTTPS protocol. Alongside the authentication module, the authorization module is implemented to ensure the privacy of end users.

**Integrity**. The HTTPS protocol adopted in the architecture helps to ensure data accuracy and consistency over its entire lifecycle. On the other hand, providing data integrity has recently become the de facto responsibility of modern databases. Thus, the proposed system is able to provide data integrity database-wise since the O-MI node supports modern well-known databases such as SQLite and H2. End users are then able to choose their own database with particular integrity features based on their own requirements.

**Availability**. Whenever required, the information must be available. This principle is provided by redundant strategies. It means that whenever the server needs to be made available all the time, the data can be replicated to several nodes and if something happens to the primary server, the data can be accessible from the other servers.

### 7.3. Security Requirements in PLIM

According to [45], PLIM systems require a security manager which can present three levels of security control including log-in access control, data access control, and functionality access control. In our model, the first level, log-in access control, is provided by the user authentication module. The other two levels, data and functionality access control, are also supported by the authorization module. Hereby, we can confirm that the proposed architecture offers the most important features mentioned in the literature. Additionally, access rules allow external users to access limited parts of the entire O-DF tree in read-only mode. Only authorized users can get access to the device’s information. Noticeably, the write access mode is restricted to a very restricted number of users. The system has publicly been running for several years and no intrusions or breaking of the access policy has been detected.

### 7.4. Attack Protection

It is impossible to log all potential intrusion attempts but the security module was tested for most possible attack scenarios, so there is no evidence indicating that the security module and policy would have been unsuccessful. According to the logs registered on the server, the server is secure against any common attack.

**Denial of service (DoS/DDoS) attack.** The network is deliberately partitioned by an attacker by transmitting invalid messages and exhausting the messages from the legitimate nodes but cryptographic schemes are not compromised. Due to receiving successive messages from the attacker, the server is unable to process the messages from the legitimate nodes. Thus, this subsequently reduces the network performance and efficiency. Compared to DoS, distributed DoS (DDoS) is a more severe type of attack, in which multiple compromised nodes from various locations run an attack on one legitimate node. We assume that the typical IoT entities have constrained resources (i.e., memory) and send the responses straight to the server. Thus, they are not prone to DoS attacks; however, DDoS is feasible and can simply be stopped by limiting the response rate for the device and request rate for users on the O-MI server.

**Man-in-the-middle (MITM) attack.** Web browser trusts the HTTPS server based on the certificate that it receives from the server containing the server’s public key. An attacker can run an MITM attack exploiting such a fact. The entire communication path will be vulnerable if this certificate is not reliable [46]. When a web client connects to a web server through an HTTPS connection, a proxy server probably comes in the middle of the connection and takes the certificate that the client has received from the web server. Since the certificate possesses the public key of the web server, the proxy cannot decrypt and modify it, and if the proxy forwards the same certificate, it will be unable to decrypt the client information. In order to run the MITM attack, the malicious proxy replaces the original certificate authenticating the HTTPS server with a modified certificate, after which it is able to decrypt the client information. However, such modification damages the signature of certification authorities, triggering a warning notification to the web client. Therefore, this attack is unsuccessful unless the user neglects to double-check the certificate.

**Replay attack.** It is a type of attack where valid data transmission is maliciously or fraudulently repeated or delayed. Entity authentication provided by HTTPS in our system is applied to ensure that the recently received message is live and fresh. It is employed to prohibit a message replay attack among IoT entities. It ascertains that the outbound and inbound messages are reasonably within a small time slot [47].

**Masquerade attack.** The malicious network entity can access the network by spoofing the actual identities of the entities leading to masquerade attacks. The attacker adopts a fake identity to gain unauthorized access to personal information through legitimate access identification. On the device side, either physically (e.g., object emulation attack) or remotely (e.g., botnet attack), an attacker can access the device ID and adopting this ID, sends a message on behalf of the original device. Given this concern, we determined various device profiling methods in [43,48], considering a combination of sensor measurements, statistical features, and header features. Accordingly, a classification-based device identification framework was developed which runs over the O-MI node (server) to fight any type of masquerade attacks.

## 8. Use Cases

To validate the proposed security architecture and verify if it supported various security requirements of users and products, the security architecture was implemented on several pilot projects of smart-city [49] and smart-building use cases. The wide set of use cases endorses the applicability of the proposed architecture.

### 8.1. Smart Building: Väre

The security architecture proposed in this research work was tested and validated in the smart-building use case of the Väre building at the Aalto University campus. The Väre building comprises 24 blocks in the whole building having a total property area of 34,000 $m^{2}$ and three floors. The design of the building considered energy efficiency with the deployment of renewable energy sources such as geothermal and solar energy. Being considered an existing physical product, connections in Väre were enabled by technologies such as Wi-Fi, sensors, O-MI reference implementation, and open messaging standards. To facilitate seamless integration of different sensors, a standard data messaging interface and data format (i.e., O-MI and O-DF) were employed for data exchange. The O-MI node server communicated with the authentication and authorization modules employing the proposed security architecture to authorize the message requests.

### 8.2. Smart Cities—bIoTope

A large amount of private and sensitive data management about civilians in smart city applications gives rise to many security challenges as it directly affects the lives of people [50]. Hence, it is vital to implement a security architecture in smart-city use cases in order to verify security metrics in these scenarios. Three pilot projects were implemented in different cities in Europe consisting of Helsinki, Lyon, and Brussels, for validating the integrated IAM system. By combining these pilot projects, an open IoT ecosystem referred to as bIoTope (https://biotope-project.eu, accessed on a January 2023) was built to empower organizations to easily create IoT systems with minimal investment and quickly manipulate the information employing systems-of-systems (SoS) capabilities for connected smart objects. This ecosystem included IT systems, web services, and functional APIs.

Based on the implemented smart-city use cases, the employed IAM approach on the client side was modular, flexible, and open source. Furthermore, the security architecture enabled the reuse of the software components by providing a common interface for all product-related data. The standard authentication module was implemented by the IoT systems with the help of modularity features using various security tools. This enabled a simpler application development by running multiple pieces of hardware and software at the same time. ML algorithms were also employed on the device side to extract features from packet headers for product identification. The automation prevented any further modifications to the products for their identification. This would result in shortening the installation and mentoring process of the product and eventually making systems growth more straightforward.

### 8.3. Prototype Implementation

To evaluate the device identification model proposed on the device-side security of the architecture, a prototype system in an office was implemented, containing six sensors, IoT gateways, routers, a virtual server, and a security module on the server. Three feature sets including measurement-based, header-based, and statistical were defined to create profiling methods for each sensor, but since sensor measurement and header features were more informative (as shown in [43]) the performance of two profiling methods, i.e., measurement-only and measurement-header, were analyzed through two attack scenarios. Each scenario included a different number of forged sensor(s) sending data in four intervals (30, 50, 70, and 100). As seen in both scenarios in Figure 11, the profiling method built upon two feature sets (i.e., measurement-header) performed better than the method based on a single feature set. Therefore, it is vital to employ sensor measurement together with the header information for device identification.

## 9. Conclusions and Future Work

Many industries have been successfully influenced by PLIM because of its compound character in both features and benefits. However, to the best of our knowledge, no security model has been proposed for PLIM that also integrates and coordinates the IoT ecosystem. To meet this challenge, the outcomes of this research are multifold. Firstly, a security architecture was proposed for PLIM based on two novel properties for security in IoT ecosystems: the product lifecycle and the distinct security requirements of products and clients; secondly, it also considered security approaches in the two domains of users and devices. On the user side, an IAM approach was proposed, which was open source, flexible, and modular, and which allowed local processing. On the device side, the device identification was managed by means of a machine learning approach using sensor measurements or features extracted from packet headers. Thirdly, the proposed architecture was validated by the use cases in different smart city scenarios. Lastly, the result analysis was performed in a prototype implemented in a real environment (i.e., an office) to validate the system. Hence, based on our findings, this is a comprehensive security architecture that can be adopted in IoT environments over the whole product lifecycle.

### 9.1. Limitations

As the security architecture provides a common interface for all product-related data, the reuse of software components is possible, which encourages rapid prototyping and innovative data usage. Furthermore, the modularity of the system allows the use of multiple standard authentication methods. As a result, many tools can be used for authentication, which eases the development of applications. On the other hand, the default access control module contains a specific permission model and interface that might not be supported by existing tools. The device-side security is mostly automated in addition to not needing any changes for the devices, thus reducing the work required to install and supervise devices, which increases the growth of the system with the same amount of work.

### 9.2. Future Work

In the future, we plan to add more security features, such as anomaly detection, to the device-side security. For this purpose, a new machine learning method will be proposed by combining the simplicity of clustering techniques and the accuracy of classification methods. Furthermore, current device-side security relies on passive authentication which can be improved by appending active authentication methods such as certificates or access tokens.

## Figures and Tables

**Figure 1 sensors-23-03236-f001:**
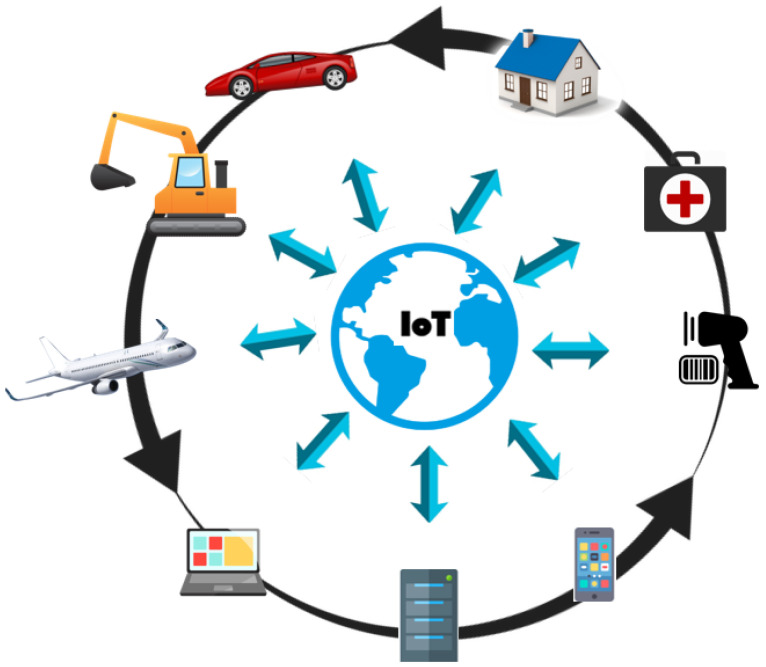
IoT interoperability.

**Figure 2 sensors-23-03236-f002:**
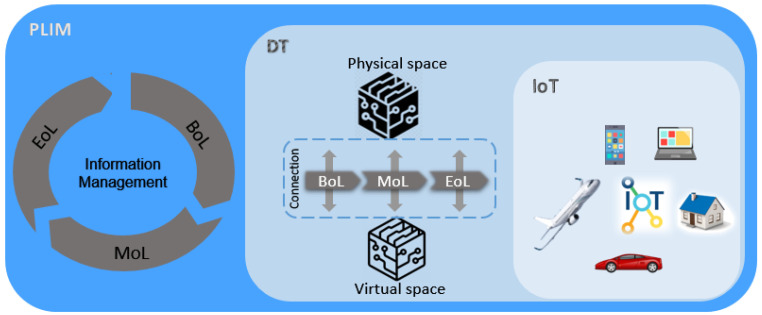
Intersection among PLIM, IoT, and DT.

**Figure 3 sensors-23-03236-f003:**
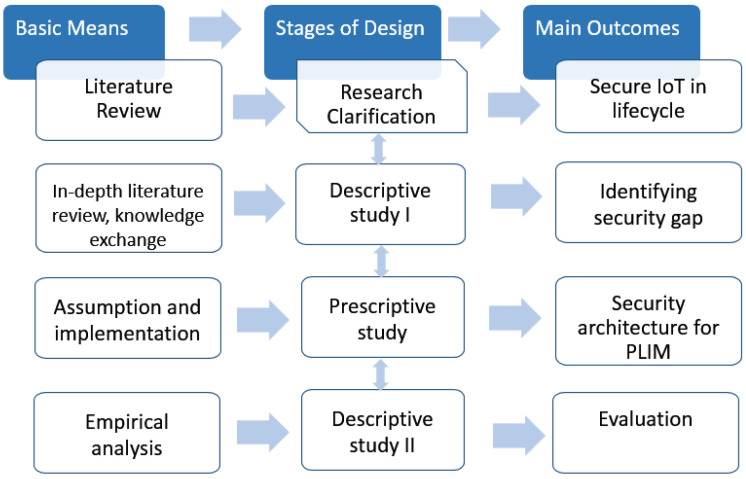
Stages of research methodology.

**Figure 4 sensors-23-03236-f004:**
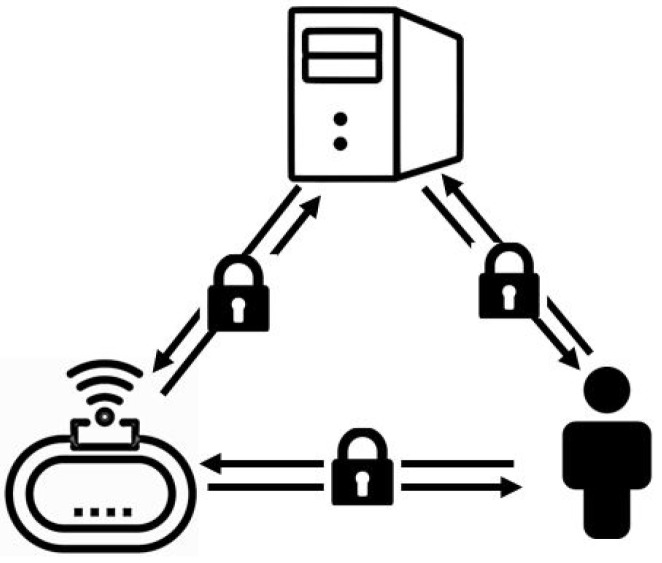
Secure IoT communication.

**Figure 5 sensors-23-03236-f005:**
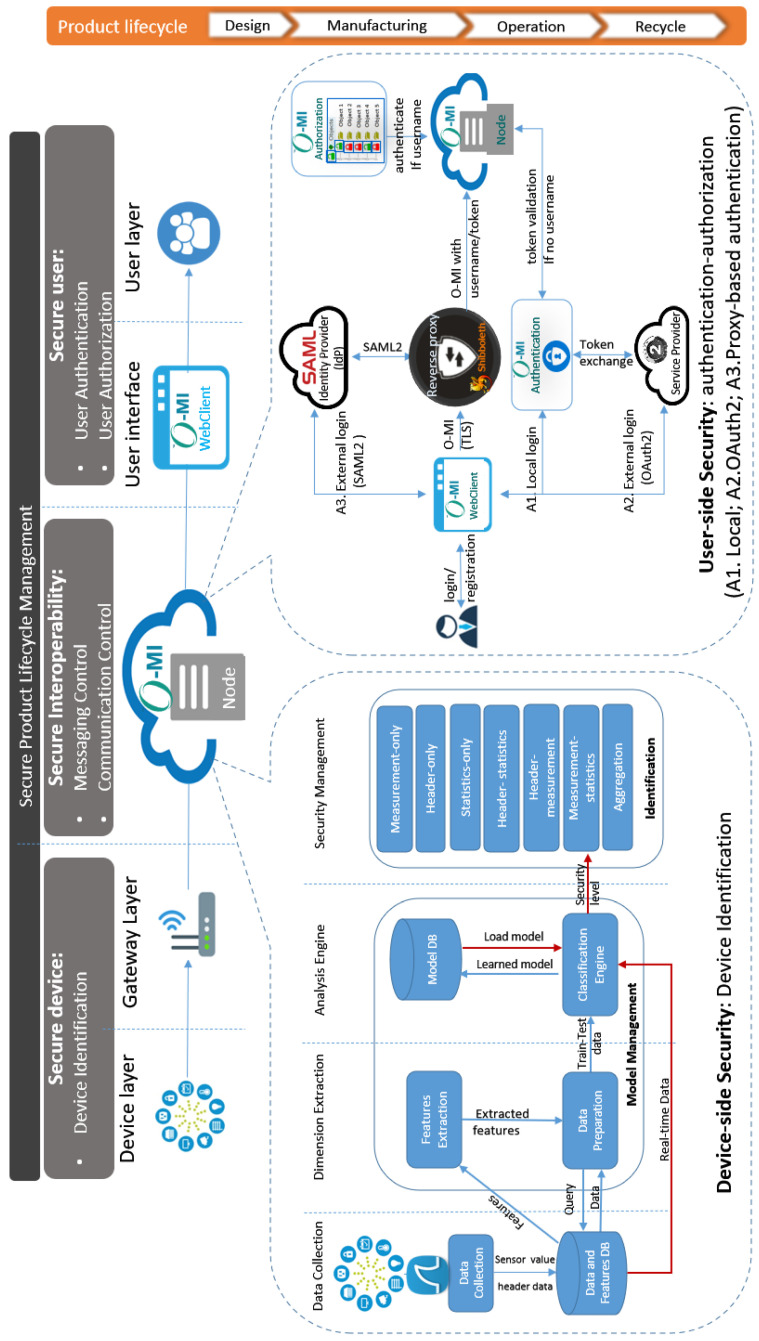
An overview of the security architecture over PLIM.

**Figure 6 sensors-23-03236-f006:**
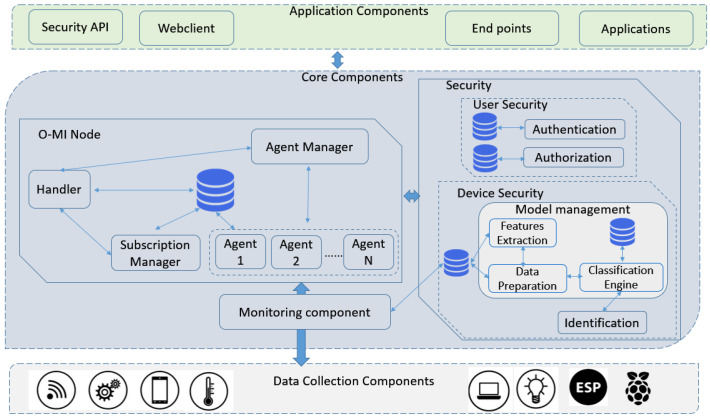
Service components over PLIM.

**Figure 7 sensors-23-03236-f007:**
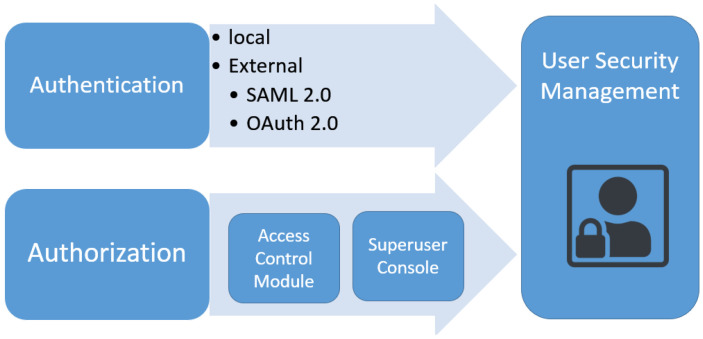
Authentication and authorization module.

**Figure 8 sensors-23-03236-f008:**
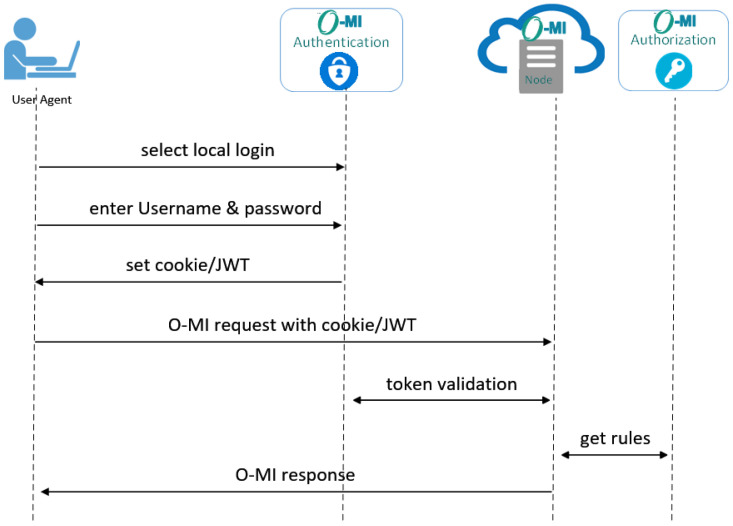
Local authentication.

**Figure 9 sensors-23-03236-f009:**
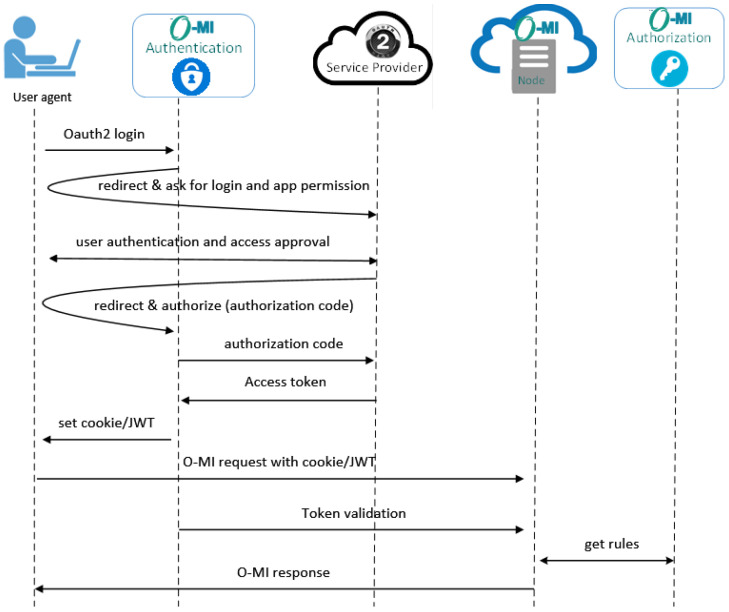
OAuth2 authentication.

**Figure 10 sensors-23-03236-f010:**
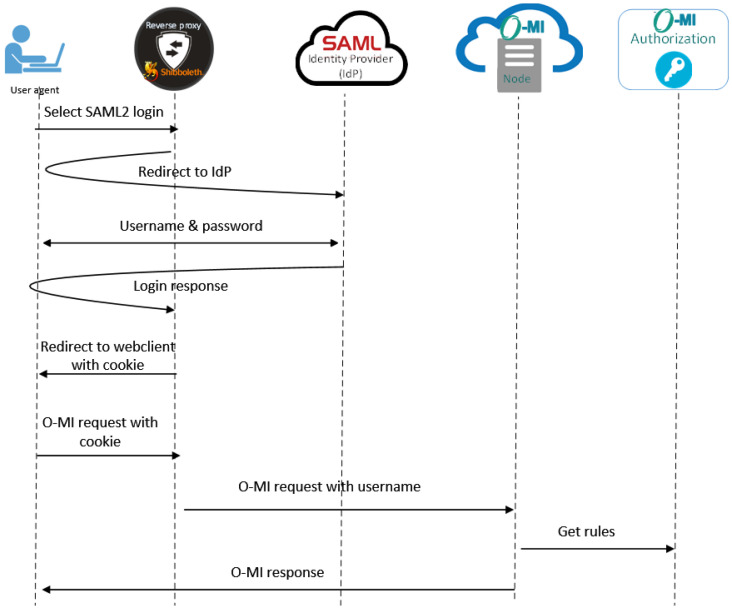
SAML authentication.

**Figure 11 sensors-23-03236-f011:**
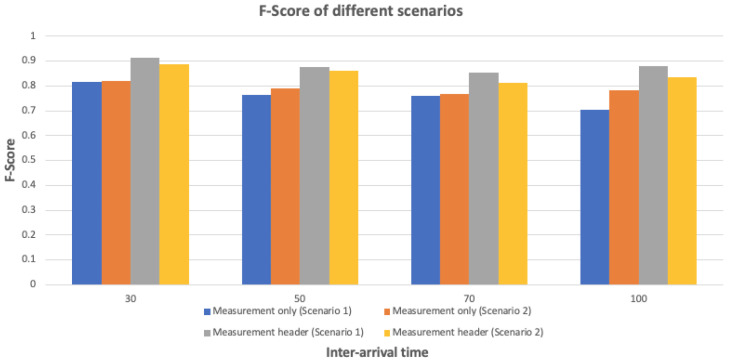
F-Score results for the prototype implemented in real environment.

**Table 1 sensors-23-03236-t001:** Comparison of different IAM tools (✓ and ✗ means that the IAM presents or not present the feature, respectively).

	AWS IAM	Google Cloud IAM	Azure IAM	Proposed IAM
Grant sharing	✓	✓	✓	✓
Granular permissions	✓	✓	✓	Tree-based
Identity federation	✓	✓	✓	✓
Single sign-On (SSO)	✓	✓	✓	✓
Web and command line access	✓	✓	✗	✓
Free access	Only Amazon customers	Only Google customers	Only Microsoft customers	✓
Open source	✗	✗	✗	✓
Flexible and modular	✗	✗	✗	✓
Local processing	✗	✗	✗	✓

**Table 2 sensors-23-03236-t002:** Classification of security issues.

Lifecycle Stage	Lifecycle Phase	Security Issue
BOL	Manufactured	Certificate installation
		Physical security
	Deployment	Identification
		Key pairing
		Vulnerability management
		Security requirements: Authentication, Authorization, Confidentiality, Integrity, Availability, Non-repudiation
MoL	Monitoring and diagnosis	Identification
		Trust
		Privacy
		Compromise detection
		Security requirements: authentication, authorization, confidentiality, integrity, availability, nonrepudiation
	Updates	Key/certificate updates
		Software updates
	Reconfiguration	Application reconfiguration
	Corporability	Mobile security
		End-to-end security
EoL	Reownership	Key/certificate update
	Decommissioned	Key/certificate revocation

## Data Availability

Not applicable.

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
