# Peer review of "A Comprehensive Security Architecture for Information Management throughout the Lifecycle of IoT Products"

_sensors, 2023, doi:10.3390/s23063236_

Round 1
Reviewer 1 Report
I think the paper is interested and some suggestions:
1 a case study such as Iot security for Vehicle for your proposal is better. you can add this part for discussion
2 some close reference can be selected for discussed or referred such as
Vehicle and Pedestrian Detection Algorithm Based on Lightweight YOLOv3-Promote and Semi-Precision Acceleration. IEEE Trans. Intell. Transp. Syst. 23(10): 19760-19771 (2022)
3 confidentiality and privacy has not discussed in your paper, please give more discussion on this.
Author Response
Regarding the comments:
- We can’t add this part to the paper, since on one hand, we have designed the generic security architecture mainly taking into consideration smart homes and smart cities and not specific to vehicles. On the other hand, the use cases mentioned in Section 7 are real implemented use cases.
- The reference has been cited in Section 3.1
- Confidentiality as one basic security pillar has been discussed in Section 7.2. Additionally, confidentiality measures are designed to prevent sensitive information from unauthorized access attempts. That’s why we have an authorization module ensuring user privacy in the framework. One sentence regarding privacy has been added to the end of Confidentiality.
Reviewer 2 Report
The manuscript addresses the problem of the security management in Product Lifecycle Information Management (PLIM) and IoT, and proposes 1) a new security architecture for PLIM and IoT and new identity and 2) access management system that could be deployed in the IoT environment.
The paper is well written and structured, it provides comprehensive background in security management, and details the methodology, which was used to identify security requirements that take place at all stages of the PLIM.
The proposed architecture was used to implement two different use cases that are different at their scale – smart building and smart city, thus demonstrating applicability and scalability of the proposed solution.
There are some minor issues to be clarified.
1) In my opinion, contribution consists in two key aspects : new security architecture for PLIM and IoT and IAM system. The other two aspects are the specific and important features of the proposed solutions. Thus, I think the contribution could be slightly rewritten.
2) The authors present a very detailed description of the research methodology, specifying all the phases, and the sources for research papers. It is interesting to know what search words were used to select research papers at each life cycle stage in order to identify the security requirements.
Author Response
Regarding the minor issues:
- The contribution part in the Introduction section has been updated and the last two aspects have been removed from the Contribution part.
- The keywords “IoT” and “security” are present in each research paper’s abstract and while searching for the specific security issues, the keywords such as BoL, MoL, and EoL were been added.
Reviewer 3 Report
This article proposes a security architecture for IoT, by particularly considering the requirements of Product Lifecycle Information Management (PLIM) against opponents which can attack the system during the whole lifecycle of an IoT 9 product.
The paper should be revised by considering the following issues:
MAJOR ISSUES
+ Introduction section should be improved to give the motivation more clearly.
+ The related work of the paper should be improved by adding more references. Some other physical layer security methods like RF fingerprinting techniques can perform very well for security attacks (like masquerade attacks, spoofing attacks) against the cyberphysical systems. What is the motivation of the proposed approach in this paper? For this purpose, I strongly recommend the authors should include the following paper in their related work in order to clarify not only the main contribution but also motivation of this paper in the related literature.
-O. M. Gul, M. Kulhandjian, B. Kantarci, A. Touazi, C. Ellement and C. D'Amours, "Fine-grained Augmentation for RF Fingerprinting under Impaired Channels," 2022 IEEE 27th International Workshop on Computer Aided Modeling and Design of Communication Links and Networks (CAMAD), Paris, France, 2022, pp. 115-120, doi: 10.1109/CAMAD55695.2022.9966888.
+ Most of the references in this paper are mostly recent publications (within the last 5 years) and relevant. On the other hand, the bibliography should be improved by adding most recent references.
+ System Model and Problem Definition should be explained more clearly as a separate section.
+ The proposed scheme performs well. The motivation behind it should be explained better.
+ Preamble information between section "6. Security Considerations" and subsection "6.1. Authentication and Authorization" should be improved.
+ The figures/schemes are generally clear. They show the data properly. It is not difficult to interpret and understand them. On the other hand, Figure 7 should be clearly explained, especially in the text/main body of the paper.
+ Numerical results should be provided to evaluate the performance of the proposed architecture.
+ Section "7. Use cases " should be improved.
+ The conclusion should be improved by giving the key results and main contributions more clearly.
MINOR ISSUES
+ The grammatical errors and typos should be fixed.
+ Sizes of Figure 1 and Figure 7 should be increased.
+ The references in the bibliography should be given in the same style. The following link should be checked: https://www.mdpi.com/authors/references
Author Response
Here is a point-by-point response to the reviewer's comments:
MAJOR ISSUES
- Introduction section should be improved to give the motivation more clearl
- One subsection named Motivation has been added to the Introduction.
- The related work of the paper should be improved by adding more references. Some other physical layer security methods like RF fingerprinting techniques can perform very well for security attacks (like masquerade attacks, spoofing attacks) against the cyberphysical systems. What is the motivation of the proposed approach in this paper? For this purpose, I strongly recommend the authors should include the following paper in their related work in order to clarify not only the main contribution but also motivation of this paper in the related literature.
- M. Gul, M. Kulhandjian, B. Kantarci, A. Touazi, C. Ellement and C. D'Amours, "Fine-grained Augmentation for RF Fingerprinting under Impaired Channels," 2022 IEEE 27th International Workshop on Computer Aided Modeling and Design of Communication Links and Networks (CAMAD), Paris, France, 2022, pp. 115-120, doi: 10.1109/CAMAD55695.2022.9966888.
- More references have been cited in Section 3.1, including the proposed paper and one paper from the same research group.
- The motivation for the proposed security architecture is partially presented at the end of Section 3.1. The motivation of the paper, in the perspective of related work is also available at the end of Section 3, named as “Conclusive Research gap”.
- Most of the references in this paper are mostly recent publications (within the last 5 years) and relevant. On the other hand, the bibliography should be improved by adding most recent
- More citations have been added to Subsection 3.1 in Related Work.
- System Model and Problem Definition should be explained more clearly as a separate section.
- A new section called “System Model and Problem Definition” has been to the paper.
- The proposed scheme performs well. The motivation behind it should be explained better.
- The motivation of the paper has been updated in the Introduction.
- Preamble information between section "6. Security Considerations" and subsection "6.1. Authentication and Authorization" should be improved.
- One sentence has been added to the beginning of Section 6.
- The figures/schemes are generally clear. They show the data properly. It is not difficult to interpret and understand them. On the other hand, Figure 7 should be clearly explained, especially in the text/main body of the paper.
- To enlarge the Figure 7, it has been separated into three figures (Figures 8, 9, 10). The explanation for each figure was available in the text without mentioning their names. Now, they have clearly been explained.
- Numerical results should be provided to evaluate the performance of the proposed architecture.
- A new subsection called Prototype Implementation (8.3)
- Section "7. Use cases " should be improved.
- Use cases have been published separately so they have been presented very briefly in the current paper. More information including the web link to the biotope project has been added.
- The conclusion should be improved by giving the key results and main contributions more clearly.
- The conclusion has been updated to include the key results and main contributions.
MINOR ISSUES
- The grammatical errors and typos should be fixed.
- It has been reviewed and type-checked.
- Sizes of Figure 1 and Figure 7 should be increased.
- The size of Figure 1 has been increased. In order to increase the size of Figure 7, we had to separate the figure into three figures: Figures 8, 9, 10
- The references in the bibliography should be given in the same style. The following link should be checked: https://www.mdpi.com/authors/references
- I am using the bibliography style in the MDPI’s template and I didn’t make any changes to the references in the template.
Round 2
Reviewer 3 Report
The authors have addressed my comments on the previous version of the paper considerably so it is acceptable for the publication in its current form.